# Study on Compression Deformation and Damage Characteristics of Pine Needle Fiber-Reinforced Concrete Using DIC

**DOI:** 10.3390/ma15051654

**Published:** 2022-02-23

**Authors:** Yonggang Wang, Shan Gao, Wei Li

**Affiliations:** 1Shaanxi Key Laboratory of Safety and Durability of Concrete Structures, Xijing University, Xi’an 710123, China; 20180004@xijing.edu.cn (Y.W.); lw18894006385@163.com (W.L.); 2School of Civil Engineering, Harbin Institute of Technology, Harbin 150090, China

**Keywords:** digital image correlation (DIC), natural fiber-reinforced concrete, pine needle fiber, crack detection and quantification, damage evolution factor, compression damage process

## Abstract

Natural fiber-reinforced concrete (NFRC) has the advantages of environmental protection, energy conservation and regeneration. However, studies conducted to improve the macro mechanical properties of concrete by pine needle fiber have achieved good results. In this paper, the deformation and compression damage of pine needle fiber-reinforced concrete (PNFRC) are analyzed by digital image correlation; a fractal dimension is used to quantify the shape of PNFRC after compression damage; and the results of scanning electron microscopy confirm the effect of fiber treatment on deformation and damage of concrete. The results showed that the horizontal strain field of PNFRC has strain concentration zones in the elastic deformation stage, indicating that the fiber enhances the deformation ability of concrete. The defined damage factor can reflect the damage of fiber-reinforced concrete (FRC). The damage curve of natural fiber concrete increases evenly and slowly compared to ordinary concrete.

## 1. Introduction

Fiber-reinforced concrete has better flexibility and strength than ordinary concrete. However, natural fiber-reinforced concrete (NFRC) has the advantages of environmental protection, energy saving and regeneration. In previous studies, natural fiber dramatically improved the macroscopic mechanical properties of concrete. The researchers observed that palm fiber [1] reduced the early drying shrinkage of concrete; it is found that the addition of natural fibers creates a good thermal insulation performance for cement mortar, hemp shives [2], flax straw, granular cork, and palm fiber [3]. Especially in terms of mechanical properties, the researchers reported that natural fibers (such as sisal fiber [4], coir [5,6], coconut [7,8], açaí fiber [9] and human hair fiber [10]) improve the compressive strength and flexural strength of NFRC. Using sisal fiber could enhance the impact resistance [11,12] of NFRC. The addition of natural fiber (coconut shell fiber) [13] improves the ductility of concrete, compared with traditional concrete. The hornification of vegetable fibers [14] has a good effect on the durability of cement mortar composites. Meanwhile, the mechanical properties of cement mortar can be significantly improved by the pretreatment of natural fibers [15,16]. Zhou et al. [17] reported that the mechanical properties of hemp fiber concrete treated by boiling and alkalization are at least 10% higher than those of plain concrete. After heat treatment and Na_2_CO_3_ treatment on the surface of sisal fiber, the durability of sisal fiber in concrete improved, as demonstrated by Wei and Meyer [18].

In particular, previous studies found that pine needle fibers [19] significantly improved the macroscopic mechanical properties of concrete. Wang, Y. and Long, W. [20] found that the descending section of the stress–strain curve of NFRC was significantly changed by pine needle fiber. However, the overall damage and deformation process of NFRC has not been considered in current research. At the same time, the failure mechanisms and processes involving concrete are the basis of concrete mechanics research and have important guiding significance for the treatment and prevention of practical engineering. For this problem, the whole process of damage and deformation of the NFRC was analyzed by digital image correlation technique. 

Digital image correlation (DIC) technology has the advantage of measuring objects without contact, which can obtain higher measurement accuracy than the traditional method [21] and can analyze the deformation information of any time and area during the loading process. Guo, Q., Wang, H. and Gao, Y. et al. [22] analyzed the effects of three kinds of mixed fibers on the deformation and strain of an asphalt mixture by DIC technology in low-temperature and crack resistance tests. The researchers measured the displacement and deformation of FRC (ultra-high performance fiber-reinforced concrete [23,24,25] and fabric-reinforced cement-based mortar [26]) and monitored the evolution of the crack strain field under direct tensile tests and bending. The changes in the crack number and crack opening of cement-based composite samples under direct tensile action at different curing ages [27] were monitored; the crack width distribution of UHPFRC and the stress crack width relationship during tension were obtained by an inverse calculation [28]. In addition, using DIC as a fracture mechanics tool, the bonding properties of carbon fiber-reinforced polymer [29] and fabric-reinforced cement-based mortar (FRCM) sheets [30] to concrete matrix were studied. Golewski, G.L. [31] proposed an analysis of the fracture process of type II concrete. Allami, K. and Colombi, P. et al. [32] studied and calculated the tensile properties of epoxy resin and CFRP sheets and the fracture energy of the CFRP concrete interface. In particular, Francisco, V. and Moro, C. et al. [33] evaluated the flexural strength, toughness, and adhesion of recycled polypropylene fiber-reinforced cementitious materials under high-temperature exposure, mainly using DIC and a scanning electron microscope (SEM). At the same time, some researchers analyzed the damage performance of concrete and rocks based on the data acquisition results of DIC [34,35,36,37,38].

In this paper, the deformation field and damage characteristics of PNFRC under compression are analyzed by DIC. All samples in this study contained a volume of 0–2% pine needle fibers treated with tap water soaking, boiling water, and diluted alkali, respectively. The time-varying evolution law of relative deformation field on the specimen surface, the initial cracking position of the specimen surface and the influence range of fiber on the deformation field were studied. Meanwhile, the failure mode and fractal characteristics of PNFRC were calculated. The influence of pine needle fibers on the deformation and damage of concrete is validated by the SEM results. Compared with different damage factor curves, the damage factor $D_{\mathrm{fa}}$ accurately describes the damage of fiber-reinforced concrete in each stress stage.

## 2. Experimental Section

### 2.1. Test Materials and Specimens

#### 2.1.1. Materials and Mix Design

In this study, the coarse aggregate used for raw materials was 5–15 mm of continuously graded gravel, and natural river sand with fineness modulus of 2.95, conforming to the Chinese Standard GBT14685-2001 and GBT14684-2001, respectively. The cementitious material was 42.5 R ordinary Portland cement; The test water met the domestic water standard GB5749-2006 [39]. The concrete mix proportion created for this experiment was 0.49:1.0:1.615:2.636 (water:cement:sand:stone), and the strength grade was 30 MPa (C30). After the pine needle fiber was processed into 30 mm, it was pretreated in three different ways. The treated procedures and effects are shown in more detail in [19].

#### 2.1.2. Specimen Design

All test samples were prepared, poured, and maintained in strict accordance with GB50081-2019 [40]. The sample size was designed as a cube of 100 × 100 × 100 mm^3^. The fiber contents added to the concrete were 0.5%, 1.0%, 1.5% and 2.0% of the volume of the test block, respectively. Table 1 shows the grouping information of test samples. In the table, the ‘T’, ‘B’ and ‘A’ represent three treatment methods (tap water soaking, boiling water, and alkalinized solution), respectively; the ‘30′ means the fiber length, and the ‘C’ stands for the compressive test. The number after C is the fiber volume content. CC is plain concrete.

### 2.2. Test Materials and Specimens

The experimental system for studying the compression deformation and damage characteristics of pine needle fiber-reinforced concrete included a loading system and DIC acquisition system.

#### 2.2.1. Digital Image Correlation (DIC)

The DIC acquisition system is shown in Figure 1a. CCD cameras (Manta G-146B/C with a Sony ICX267 sensor, dpi: 2452 × 2056, Allied Vision, Puchheim, Germany) were used in this research, which ran at 10 fps(full resolution). The CCD cameras, using DIC technology, obtained the camera parameters through correction (Figure 1b) and collected the strain and deformation of speckle samples (Figure 1c) during the experiment. The strain region of the compressive damage process analysis and statistical damage analysis is shown in Figure 1d, and the linear strain region of this is displayed in Figure 1e. The software MATCHID-2D/STEREO Correlated 2018 (University of Leuven, Leuven, BE) was used to carry out the DIC analyses.

#### 2.2.2. Compressive Strength Test of PNFRC

Hydraulic servo universal testing machine (MTS.e64.206, MTS Systems Corporation, Eden Prairie, MN, USA) was selected for the loading system, as shown in Figure 1a. According to GB50010-2010, all specimens were tested after 28 days of standard curing. The loading rate of the pressure testing machine was 0.5 MPa/s, and the data acquisition frequency of the pressure testing machine was set to 2 Hz.

## 3. Results and Analysis

### 3.1. Cracking Mode and Fractal Characteristics of PNFRC Samples

Figure 2 illustrates the main cracks of the PNFRC samples after peak loading, and the blue color indicates the main cracks and the concrete shedding part of the concrete. Clearly, the PNFRC treated with alkali show many small cracks after rupture, which prove that the deformation of concrete is restrained and has a bridging effect between the concrete and pine needle fibers; the tap-water-treated pine needle fiber-reinforced concrete show mainly block shedding after damage, which indicates the inhibiting effect of tap-water-treated fibers on the cement hydration of concrete; however, the damage pattern of boiling-water-treated pine needle fiber-reinforced concrete is between the alkali-treated and tap-water-treated PNFRC.

The fractal dimension is used to evaluate the failure morphology of concrete with three treatments. Figure 3 shows the fractal dimension of cracks of pine needle fiber-reinforced concrete specimens. In general, the fractal dimension of alkali-treated PNFRC is slighter than that of the CC specimen, while the fractal dimension of boiling-water-treated and tap-water-treated fiber concrete is much greater than that of the plain concrete, which is consistent with the macro results of a damaged morphology.

### 3.2. Micro-Analysis of SEM

From the micro-level, the effects of different treatments of pine needle fibers on the deformation of concrete are demonstrated by SEM, as shown in Figure 4. In general, there are varying degrees of cement mortar on the fiber surface compared to the original pine needle fiber (Figure 4a). In particular, the surface of the fiber treated by dilute alkali is wrapped with the most cement mortar (Figure 4b), and the fiber adhere well to the cement matrix, cracking the cement matrix after destruction and particularly influencing the deformation and damage of the concrete.

### 3.3. The Compressive Damage Process Analysis

Figure 5 displays the load-displacement curves of all samples. Clearly, the content and treatment of pine needle fibers have a direct influence on the macro mechanical properties and deformability of concrete. The effect of pine needle fibers treated with boiling water and diluted alkali on concrete is greater than PNFRC treated with tap water in strength, deformation peak value, and elastic modulus. Therefore, the horizontal strain field of pine needle fiber concrete treated with boiling water and dilute alkali was selected for analysis. The horizontal strain field corresponding to point A–E of the load-displacement curve of concrete samples was mainly analyzed. Point A–E is the corresponding load value of 10%, 50%, 70%, 90% and 100% of the peak value (Pmax ), respectively, as shown in Figure 6 and Figure 7.

At the early phase of loading (before A), the coordinated deformation of the corner part of the specimen and the indenter resulted in local breakage, causing a weak strain concentration. In the early loading stage A–C, the strain field of the CC sample is relatively uniform, as the samples are in the linear stage of the load-displacement curve under compression. However, the strain fields on the surface of the PNFRC specimen present an uneven shape. At the elastic stage, the relative deformation fields of PNFRC were formed, and fiber and concrete bear forces together.

In the C–D stage, there are strain concentration areas on the surface of the specimens in the nonlinear part of the load-displacement curve, and the strain field shows the clear heterogeneity of zoning. Especially for the specimens containing 0.5–1.5%, the impact range of the fibers on the relative deformation field of concrete increases, and a large strain concentration area appears on the sample surface.

### 3.4. Statistical Analysis of PNFRC Damage

DIC is an important tool for quantifying sample damage. S.P. Ma [37] proposed to use the standard deviation of the horizontal strain field to represent strain field damage degree factor $D_{\mathrm{fa}}$, as shown in Equations (1)–(2). Similarly, Y.R. Zhao et al. [35] and H. Zhang et al. [38] proposed the deviation of the strain field to calculate the damage degree factor. According to a horizontal strain field analysis of compressive damage, five kinds of concrete samples with good modification effects were selected for statistical damage analysis. The damage factors, $D_{\mathrm{fm}}$ and $D_{\mathrm{fl}}$; in line strain region; and plane strain region (as shown in Figure 1d,e) were calculated by combining Equations (1) and (2). At the same time, the damage factor $D_{\mathrm{fa}}$ was defined in the linear strain region as shown in Equation (3).
(1)$$D_{f,j}=\frac{S_{j}}{S_{\max}}$$
(2)$$S_{j}=\sqrt{\frac{1}{n-1}\overset{n}{\underset{i=1}{\sum }}{\left(\varepsilon_{i}-\;\bar{\varepsilon }\;\right)}^{2}}$$
(3)$$D_{\mathrm{fa},j}=\overset{n}{\underset{i=1}{\sum }}\varepsilon_{i}/\overset{n}{\underset{i=1}{\sum }}\varepsilon_{\max}$$
where n is the number of strain points in every DIC horizontal strain field analysis region; $\varepsilon_{i}$ stands for the strain at any point in the j-th horizontal strain field; $\bar{\varepsilon }$ indicates the strain mean of the j-th horizontal strain field; and $\varepsilon_{\max}$ means the strain value in the strain field analysis region at the peak load.

Figure 8 lists the damage evolution curve of PNFRC samples with the changes in displacement. It is evident that the change of all damage factors ($D_{\mathrm{fl}}$, $D_{\mathrm{fm}}$ and $D_{\mathrm{fa}}$) of ordinary concrete samples almost coincide. In general, the changes of $D_{\mathrm{fl}}\;$and $D_{\mathrm{fa}}$ are more sensitive than those of $D_{\mathrm{fm}}$ and can better reflect the details of the damage and deformation of samples. The $D_{\mathrm{fl}}$ and $D_{\mathrm{fa}}$ are not smooth in the whole change stage, which indicates that the damage of pine needle fiber-reinforced concrete is clearly affected. The $D_{\mathrm{fl}}\;$ and $D_{\mathrm{fa}}$ of the B30C05 samples also have a mutation at the point of load mutation.

Under a 10% load at the initial loading phase, the above three damage factors keep changing synchronously. Under the same deformation conditions, the more fiber content, the earlier the damage occurs to pine needle fiber-reinforced concrete specimens. The damage degree of diluted-alkali-treated samples is less than that of the boiling-water-treated sample when the load is in the elastic stage. The increase in the damage of PNFRC is relatively gentle compared to ordinary concrete.

## 4. Conclusions

In this study, DIC technology was used to study the compressive damage characteristics of PNFRC, and three damage evaluation methods were compared. The conclusions were as follows:(1)The fractal dimension of fiber-reinforced concrete treated with tap water soaking, boiling water, and an alkali solution is consistent with the macroscopic results of damaged morphology. At the same time, it is proved that pine needle fiber can promote the deformation and damage of concrete from the micro point of view. The SEM results show that the surface of the alkali-treated fiber is wrapped with cement mortar, which produces cracks after the concrete is damaged.(2)At 50% of the peak load, the strain concentration zone of the 0.5–1.5% PNFRC appears, and the strain concentration zone develops to the maximum at 90% of the peak load. The “X” type strain concentration zone of other samples is irregular at the peak load, compared with ordinary concrete and 2% pine needle fiber concrete.(3)Adding natural fiber can change the damage growth rate of concrete and make the damage curve grow evenly and slowly. The samples A30C10 show the best deformation and damage performance. The damage degree of the samples treated with boiling water is greater than that of the samples treated with dilute alkali in the elastic deformation stage.(4)The damage factor D_fa_ could accurately reflect the compression damage of PNFRC. In the process of compression deformation, the defined damage factor can characterize the details of the damage and deformation of each specimen.

Therefore, it can be concluded that the damage of NFRC is different from that of ordinary concrete, and the damage factor evaluation method proposed in this study can accurately recognize the evolution mechanism of the plant fiber concrete instability precursor response crisis, and realize the damage evaluation of concrete. As a new type of natural reinforced composite material applied in buildings, PNFRC still needs further research. PNFRC can also be used to control the degree of damage by manual intervention.

## Figures and Tables

**Figure 1 materials-15-01654-f001:**
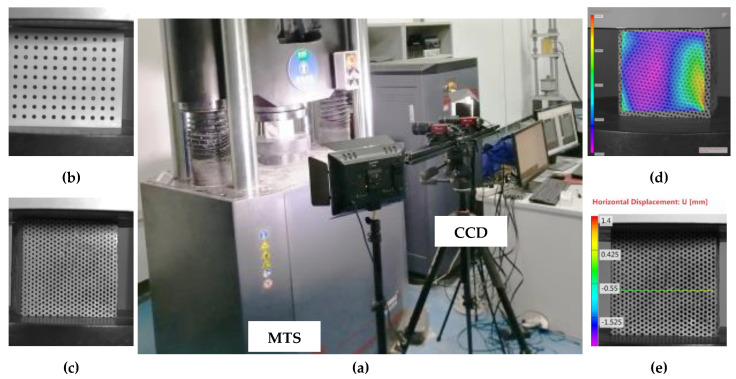
Test setup (**a**) MTS pressure testing machine and industrial camera test system (**b**) Camera parameter correction (**c**) Sample speckle (**d**) Area of surface analysis (**e**) Area of line analysis.

**Figure 2 materials-15-01654-f002:**
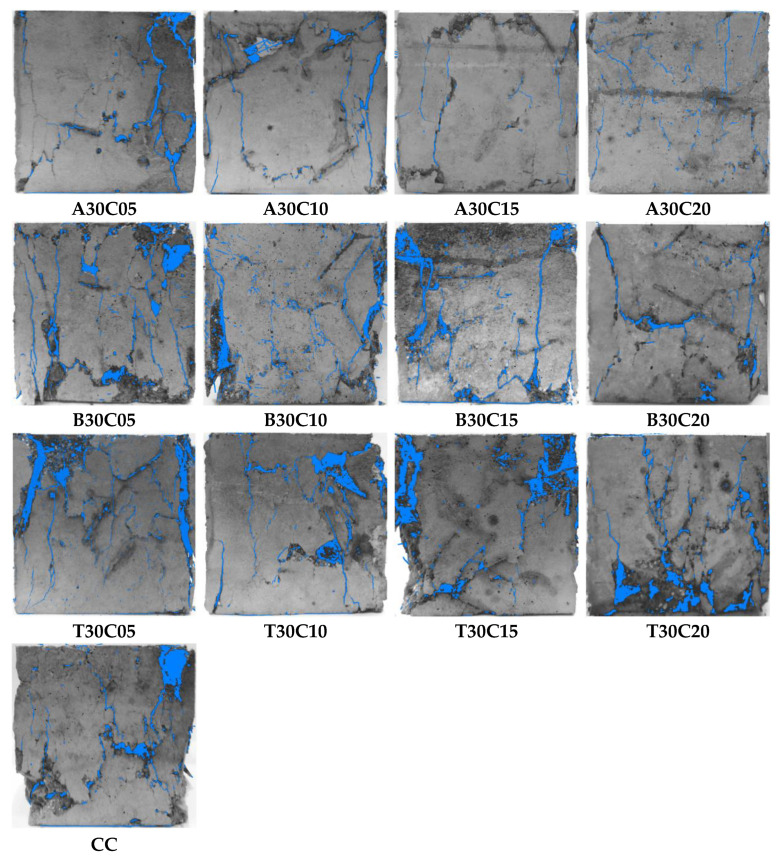
Failure characteristics of the samples.

**Figure 3 materials-15-01654-f003:**
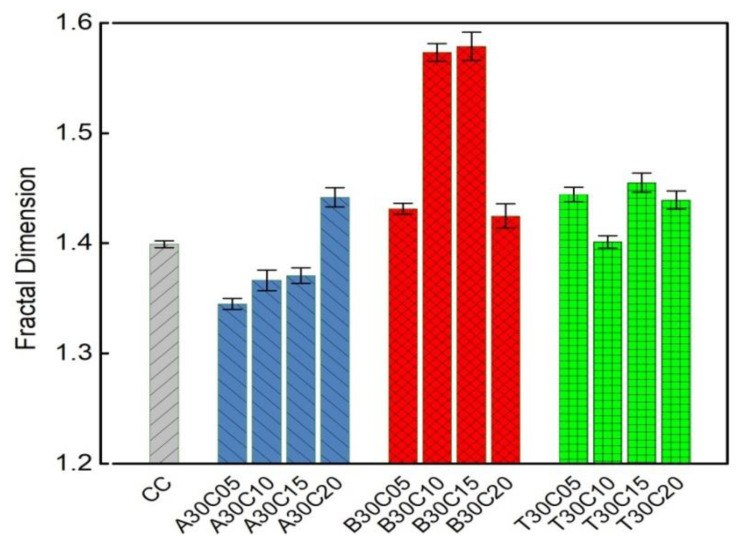
Failure characteristics of the samples.

**Figure 4 materials-15-01654-f004:**
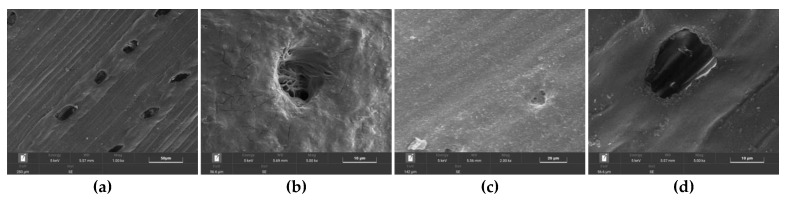
The morphology of fiber surface: (**a**) primitive fiber, (**b**) alkali-treated fiber, (**c**) boiling-water-treated fiber, and (**d**) tap-water-soaked fiber schemes follow the same formatting.

**Figure 5 materials-15-01654-f005:**
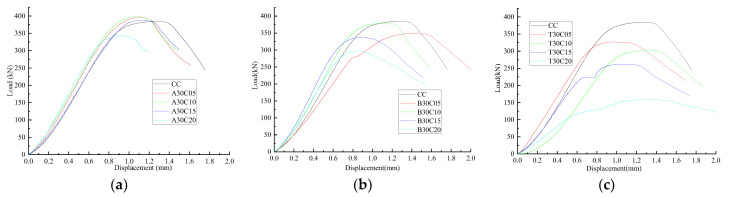
Load-displacement curves of all samples. (**a**) Alkalinized solution, (**b**) Boiling water, and (**c**) Tap water.

**Figure 6 materials-15-01654-f006:**
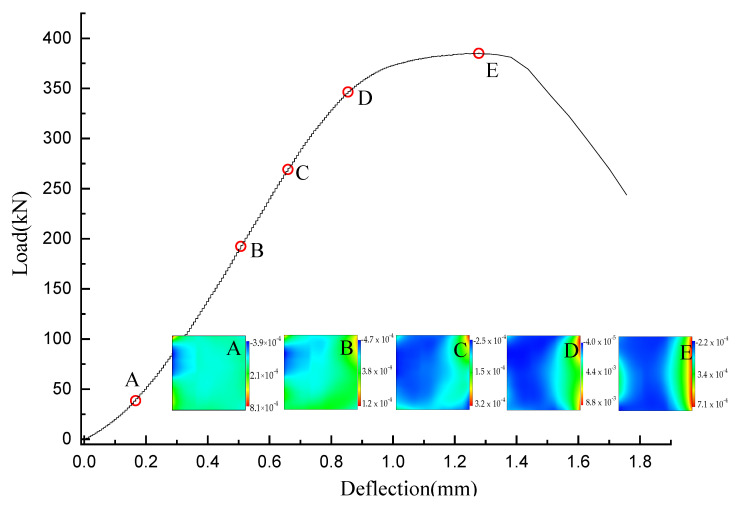
The load-displacement curve and horizontal strain field of ordinary concrete (CC) sample.

**Figure 7 materials-15-01654-f007:**
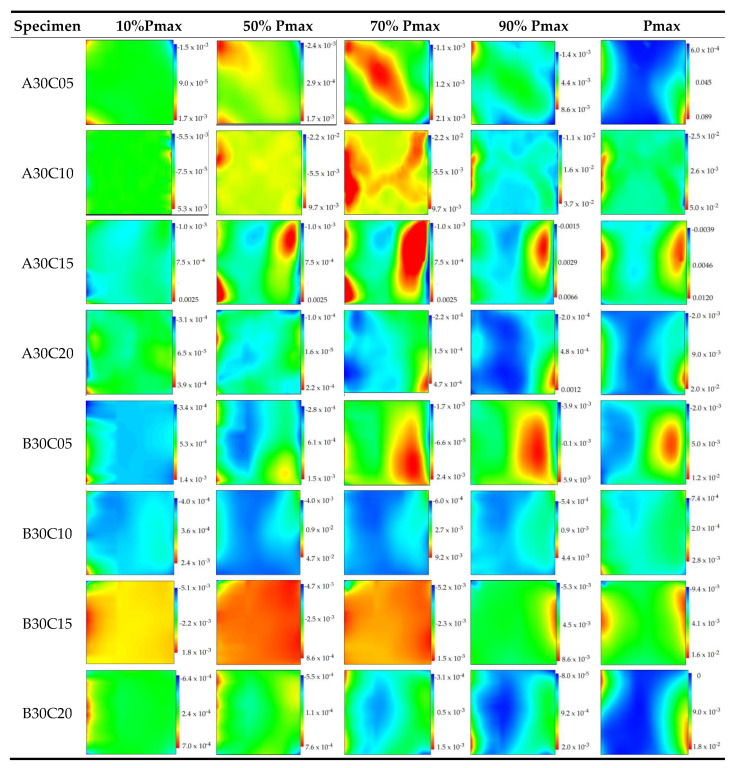
Evolution of horizontal strain field of PNFRC specimens.

**Figure 8 materials-15-01654-f008:**
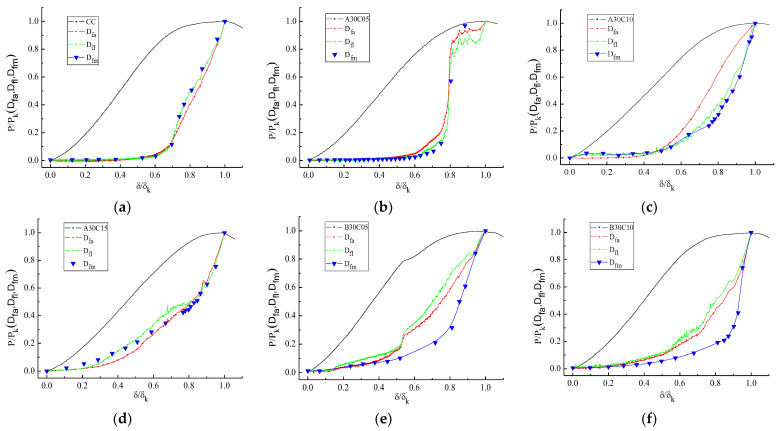
Damage evolution curve of PNFRC samples. (**a**) CC, (**b**) A30C05, (**c**) A30C10, (**d**) A30C15, (**e**) B30C05, and (**f**) B30C10.

**Table 1 materials-15-01654-t001:** Test sample details.

W/B	Sample Size	Fiber Pretreated Method	Fiber Volume				
			0%	0.5%	1.0%	1.5%	2.0%
0.49	100 × 100 × 100 mm^3^	Tap water soaking	CC	T30C05	T30C10	T30C15	T30C20
		Boiling water		B30C05	B30C10	B30C15	B30C20
		Alkalinized solution		A30C05	A30C10	A30C15	A30C20

## Data Availability

The raw/processed data required to reproduce these findings cannot be shared at this time as they form part of an ongoing study.

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
