# Peer review of "Study on Compression Deformation and Damage Characteristics of Pine Needle Fiber-Reinforced Concrete Using DIC"

_materials, 2022, doi:10.3390/ma15051654_

Round 1

Reviewer 1 Report

The research work on “Study on compression deformation and damage characteristics of pine needle fiber reinforced concrete using digital image correlation” is good, but few improvements should be made before its acceptance recommendations. Following aspects need attention:

  1. No paper is being cited from 2021 and 2022. Introduction section should be strengthened with latest literature.
  2. Avoid 1-3 sentences paragraphs, e.g. lines 25-27.
  3. Any heading should start from text instead of figure, e.g. see headings 2.1, 3.1, l.2 etc.
  4. How many samples are being tested for a single combination?
  5. Figure 5: curve for T10 may be reviewed again.
  6. Scientific reasoning emphasis is required while elaborating results.
  7. There should be a separate section explaining the implementation of this research in real field for practicing professionals/engineers/designers.
  8. There should be closing remarks at the end of conclusions.

Author Response

Dear Reviewer,

We thank you for the thoughtful review and comments on our study. We have revised the manuscript very carefully. It should be noted that the originally existed words are in black color, the writing modified words are highlighted in pink color. The following modifications are listed in this response letter.

Point 1: No paper is being cited from 2021 and 2022. Introduction section should be strengthened with latest literature.

Response 1: We added the latested references , such Ref. 17,18,25,27 and 30, in the manuscript.

Point 2: Avoid 1-3 sentences paragraphs, e.g. lines 25-27.

Response 2: We made relevant changes. The more detailed information has been marked in the section 1 in the manuscript.

Point 3: Any heading should start from text instead of figure, e.g. see headings 2.1, 3.1, l.2 etc.

Response 3: We made relevant changes.

Point 4: How many samples are being tested for a single combination?

Response 4: A total of 13 groups were designed and 3 samples were tested in each group.

Point 5: Figure 5: curve for T10 may be reviewed again.

Response 5: We made relevant changes in the all Figures.

Point 6: Scientific reasoning emphasis is required while elaborating results.

Response 6: The corresponding changes were made in section 3.1.

Point 7: There should be a separate section explaining the implementation of this research in real field for practicing professionals/engineers/designers.

Response 7: The implementation of this research in real field is added at the end of the conclusions.

Point 8: There should be closing remarks at the end of conclusions.

Response 8: The closing remarks are added at the end of the conclusions.

We look forward to your response to the changes we have made. We thank the reviewer and remain at your disposal for any further questions.  

Yours sincerely,

Yonggang Wang

Reviewer 2 Report

In article was intended to explore the natural fiber reinforced concrete (NFRC) has the advantages of environmental protection, energy conservation and regeneration. However, the improvement of macro mechanical properties of concrete by pine needle fiber has achieved good results. In this dissertation, the deformation and compression damage of pine needle fiber reinforced concrete (PNFRC) are analyzed by digital image correlation; fractal dimension was used to quantify the shape of PNFRC after compression damage; the results of scanning electron microscope confirm the effect of fiber treatment on deformation and damage of concrete. The text was prepared enough carefully but in my opinion in a lot place must was improved. I have some questions of the presented process, too. Below the suggested my corrections and my questions was presented.

Thank you for an interesting paper of an actual topic.

Corrections:

  • Language: The English language in the paper should be checked by a professional
  • Please reduce the title by using some innovative keywords for the title and also the title it is too long for this study
  • The authors use different abbreviations at different places, which confused the reader, Please provide the list of the abbreviation, please use at the start.
  • The introduction needs to be more emphasized on the research work with a detailed explanation of the whole process considering past, present, and future scope.
  • A lot of researchers have a lot of microscopic features for concrete such as Petrounias et al.2020 and Petrounias et al.2021 in sustainability journal when emphasized in the natural hair fibers. If you want you can add some references in the very interesting introduction part.
  • Section descriptions should be written as a separate paragraph at the end of the Introduction, by including the sources of material and findings of this work.
  • Research gaps should be highlighted more clearly and future applications of this study should be added. Please make the link between the advanced images used in this paper with the study on compression deformation and damage characteristics of pine needle fiber reinforced concrete.
  • Please describe the importance and novelty of the selected problem, data details. Please also provide the comparative analysis with the other available material in the international /or China market.
  • In the part 3, do you have any more microscopic information with polarized microscope? Any structure properties?
  • Figures are generated through different software or taken from the different means. Please use the same tool for producing the figures. This will improve the presentation of the paper and also ease of reading for the reader.
  • The compressive damage process analysis measurement is described in detail. On the other hand you did not even mention the method used for determining setting. Please stay to a uniform description regarding details.
  • In my opinion I need more discussion or analysis part because it is very prolix and in this form and without any useful analysis for other researchers.
  • The conclusion part is a little bit prolix and should be improved. It seems to be a sum of results. Also, limitations, future scope, and recommendations of this study are suggested to be written as a separate section. What was the research significance of this study?
  • In some parts of the manuscript, authors discussed the results in light of the literature prior to presenting their results, which is distracting.
  • Please kindly make revisions to the language of the paper presentation. There are still some minor typos and grammatical errors.

Summary:

Overall, the research method and analysis of results are well-documented and make sense. The authors presented modest introduction literature overview, too. Generally in my opinion this study it is too short for the materials journal. The subject and the methodology applied seem to be justified and deserve publication but after major reviewers. The topic is very important for the study on compression deformation and damage characteristics of pine needle fiber reinforced concrete using digital image correlation.

Author Response

Dear Reviewer,

First of all, thanks very much for your relevant comments on our article. We tried our best to improve the manuscript and made some changes in the manuscript.

Point 1: Please reduce the title by using some innovative keywords for the title and also the title it is too long for this study.

Response 1: We made relevant changes. The more detailed information has been marked in the title.

Point 2: The authors use different abbreviations at different places, which confused the reader, Please provide the list of the abbreviation, please use at the start.

Response 2: The main abbreviations are the serial numbers of test samples, and all abbreviations have been modified uniformly.

Point 3: The introduction needs to be more emphasized on the research work with a detailed explanation of the whole process considering past, present, and future scope. A lot of researchers have a lot of microscopic features for concrete such as Petrouniaset al.2020 and Petrounias et al.2021 in sustainability journal when emphasized in the natural hair fibers. If you want you can add some references in the very interesting introduction part.

Response 3: We added the latested references , such Ref. Petrounias et al.2021, in the manuscript.

Point 4: Section descriptions should be written as a separate paragraph at the end of the Introduction, by including the sources of material and findings of this work.

Response 4: We made relevant changes at the end of the introduction, including pine needle fiber materials and the damage of evaluation methods proposed in this study.

Point 5: Research gaps should be highlighted more clearly and future applications of this study should be added. Please make the link between the advanced images used in this paper with the study on compression deformation and damage characteristics of pine needle fiber reinforced concrete.

Response 5: The damage factors proposed in the study are obtained by analyzing advanced images, and the images at each moment correspond to the compression deformation of concrete, respectively.

Point 6: Please describe the importance and novelty of the selected problem, data details. Please also provide the comparative analysis with the other available material in the international /or China market.

Response 6: Fiber-reinforced concrete has better flexibility and strength than ordinary concrete. whatsmore, Natural fiber-reinforced concrete (NFRC) have the advantages of environmental protection, energy saving and regeneration. In this paper, the advantages of pine needle fiber concrete are explained in the introduction. The mechanical properties of pine needle fiber are described in detail in Ref. 17 and 18. The ordinary concrete is set up, compared to pine needle fiber reinforced concrete in this research.

Point 7: In the part 3, do you have any more microscopic information with polarized microscope? Any structure properties?

Response 7: In the part 3, the microscopic informations are used to verify the influence of different fiber treatment methods on concrete damage. However, there is more microscopic information in our previously published literature 17.

Point 8: Figures are generated through different software or taken from the different means. Please use the same tool for producing the figures. This will improve the presentation of the paper and also ease of reading for the reader.

Response 8: The software of ORIGIN was used to edit all the graphics and modify the sample numbers of all the graphics uniformly.

Point 9: The compressive damage process analysis measurement is described in detail. On the other hand you did not even mention the method used for determining setting. Please stay to a uniform description regarding details.

Response 9: The damage process analysis method is applied in Figure 8 of Section 3.3.

Point 10: In my opinion I need more discussion or analysis part because it is very prolix and in this form and without any useful analysis for other researchers.

Response 10: The main purpose of this study is to propose an evaluation method suitable for plant fiber concrete compression damage.

Point 11: The conclusion part is a little bit prolix and should be improved. It seems to be a sum of results. Also, limitations, future scope, and recommendations of this study are suggested to be written as a separate section. What was the research significance of this study?.

Response 11: The conclusion is a summary of the experimental results. Also, limitations, future scope, or recommendations of this research in real field is added at the end of the conclusions.

Point 12: In some parts of the manuscript, authors discussed the results in light of the literature prior to presenting their results, which is distracting.

Response 12: We made relevant changes.

Point 13: Please kindly make revisions to the language of the paper presentation. There are still some minor typos and grammatical errors.

Response 13: We have revised the language of the manuscript.

We look forward to your response to the changes we have made. We thank the reviewer and remain at your disposal for any further questions.

 Yours sincerely,

Yonggang Wang

Reviewer 3 Report

This paper is an interesting paper about Title: Study on compression deformation and damage characteristics of pine needle fiber reinforced concrete using digital image correlation. the deformation field and damage characteristics of PNFRC under compression are analyzed by DIC. The paper is well structured, and you need some corrections to improve the English written.

The paper can be published it these minor revisions are made.

Some comment:

  • The fibre content was add in volume or weight (ligne 75)
  • Section 2.1: Can you give the granular distribution of gravel ? what this the type of cement CEMI or II
  • why did you only test compression test in mechanical properties? i think the bending test is the most suitable to identify the performance of the concrete due to the addition of the fibres?
  • To see the effect of the treated fibres you should perform pull-out tests

  • Can you add this references
    • Mahfoud Benzerzour, Nassim Sebaibi, Nor Edine Abriak, Christophe Binetruy, Waste fibre–cement matrix bond characteristics improved by using silane-treated fibres, Construction and Building Materials, Volume 37, 2012, Pages 1-6
    • Bui, Huyen, Nassim Sebaibi, Mohamed Boutouil, and Daniel Levacher. 2020. "Determination and Review of Physical and Mechanical Properties of Raw and Treated Coconut Fibers for Their Recycling in Construction Materials" Fibers8, no. 6: 37. https://doi.org/10.3390/fib8060037
  • I think, you can determine de ductility to see the performance of fibres reinforced concrete

Author Response

Dear Reviewer,

First of all, thanks very much for your relevant comments on our article. We tried our best to improve the manuscript and made some changes in the manuscript.

Point 1: The fibre content was add in volume or weight (ligne 75)

Response 1: We made relevant changes. The fibre content was add in volume.

Point 2: Section 2.1: Can you give the granular distribution of gravel ? what this the type of cement CEMI or II

Response 2: We made relevant changes. The Coarse aggregate and fine aggregate used crushed granitic rocks and river sand, respectively, conforming Chinese Standard GBT14685-2001 and GBT14684-2001.

Point 3: why did you only test compression test in mechanical properties? i think the bending test is the most suitable to identify the performance of the concrete due to the addition of the fibres? To see the effect of the treated fibres you should perform pull-out tests.

Response 3: Thank you very much for your suggestion. We have carried out part of the ductility study in reference 17. In order to further study and wide application, the main purpose of this study is to research the compressive damage of plant fiber concrete.

Point 4: Can you add this references

Mahfoud Benzerzour, Nassim Sebaibi, Nor Edine Abriak, Christophe Binetruy, Waste fibre–cement matrix bond characteristics improved by using silane-treated fibres, Construction and Building Materials, Volume 37, 2012, Pages 1-6

Bui, Huyen, Nassim Sebaibi, Mohamed Boutouil, and Daniel Levacher. 2020. "Determination and Review of Physical and Mechanical Properties of Raw and Treated Coconut Fibers for Their Recycling in Construction Materials" Fibers8, no. 6: 37. https://doi.org/10.3390/fib8060037

Nassim Sebaibi, Mahfoud Benzerzour, Nor-Edine Abriak, Christophe Binetruy, Mechanical and physical properties of a cement matrix through the recycling of thermoset composites, Construction and Building Materials, Volume 34, 2012, Pages 226-235,

Response 4: We added the latested references , such Ref. 38, in the manuscript.

We look forward to your response to the changes we have made. We thank the reviewer and remain at your disposal for any further questions.

Yours sincerely,

Yonggang Wang

Round 2

Reviewer 2 Report

The authors have worked hard to improve and restructure this work. The authors have responded to all comments made by the Reviewer. I propose to the editors to accept this paper for publication in Materials.

Author Response

Thank the reviewer again for your recognition of our research.